# TMPRSS2 and furin are both essential for proteolytic activation of SARS-CoV-2 in human airway cells

Dorothea Bestle[1,*], Miriam Ruth Heindl[1,*], Hannah Limburg[1,*], Thuy Van Lam van[2], Oliver Pilgram[2], Hong Moulton[3], David A Stein[3], Kornelia Hardes[2,4], Markus Eickmann[1,5], Olga Dolnik[1,5], Cornelius Rohde[1,5], Hans-Dieter Klenk[1], Wolfgang Garten[1], Torsten Steinmetzer[2], Eva Böttcher-Friebertshäuser[1]

**The novel emerged SARS-CoV-2 has rapidly spread around the world causing acute infection of the respiratory tract (COVID-19) that can result in severe disease and lethality. For SARS-CoV-2 to enter cells, its surface glycoprotein spike (S) must be cleaved at two different sites by host cell proteases, which therefore represent potential drug targets. In the present study, we show that S can be cleaved by the proprotein convertase furin at the S1/S2 site and the transmembrane serine protease 2 (TMPRSS2) at the S2′ site. We demonstrate that TMPRSS2 is essential for activation of SARS-CoV-2 S in Calu-3 human airway epithelial cells through antisense-mediated knockdown of TMPRSS2 expression. Furthermore, SARS-CoV-2 replication was also strongly inhibited by the synthetic furin inhibitor MI-1851 in human airway cells. In contrast, inhibition of endosomal cathepsins by E64d did not affect virus replication. Combining various TMPRSS2 inhibitors with furin inhibitor MI-1851 produced more potent antiviral activity against SARS-CoV-2 than an equimolar amount of any single serine protease inhibitor. Therefore, this approach has considerable therapeutic potential for treatment of COVID-19.**

## Introduction

In December 2019, a new coronavirus (CoV) emerged and has rapidly spread around the world causing a pandemic never before observed with these viruses. The virus was identified as a new member of the lineage b of the genus *Betacoronavirus*, which also contains the 2002 severe acute respiratory syndrome (SARS)-CoV, and was named SARS-CoV-2 by the World Health Organization. The respiratory disease caused by the virus was designated as coronavirus disease 2019 (COVID-19).

CoVs are a large family of enveloped, single-stranded positive-sense RNA viruses belonging to the order *Nidovirales* and infect a broad range of mammalian and avian species, causing respiratory or enteric diseases. CoVs have a major surface protein, the spike (S) protein, which initiates infection by receptor binding and fusion of the viral lipid envelope with cellular membranes. Like fusion proteins of many other viruses, the S protein is activated by cellular proteases. Activation of CoV S is a complex process that requires proteolytic cleavage of S at two distinct sites, S1/S2 and S2′ (Fig 1), generating the subunits S1 and S2 that remain non-covalently linked (1, 2, 3). The S1 subunit contains the receptor binding domain, whereas the S2 subunit is membrane-anchored and harbors the fusion machinery. Cleavage at the S2′ site, located immediately upstream of the hydrophobic fusion peptide, has been proposed to trigger the membrane fusion activity of S (4, 5). In contrast, the relevance of S cleavage at the S1/S2 site is not yet fully understood. Processing of CoV S is believed to occur sequentially, with cleavage at the S1/S2 site occurring first and subsequent cleavage at S2′. Cleavage at the S1/S2 site may be crucial for conformational changes required for receptor binding and/or subsequent exposure of the S2′ site to host proteases at the stage of virus entry (reviewed in references 6, 7, and 8).

Many proteases have been found to activate CoVs in vitro, including furin, cathepsin L, and trypsin-like serine proteases such as the transmembrane serine protease 2 (TMPRSS2), TMPRSS11A, and TMPRSS11D (reviewed in references 6, 7, and 8). Among them, TMPRSS2 and furin play major roles in proteolytic activation of a broad range of viruses (reviewed in references 9, 10, and 11). TMPRSS2 is a type II transmembrane serine protease (TTSP) that is widely expressed in epithelial cells of the respiratory, gastrointestinal, and urogenital tract (11, 12). The physiological role of TMPRSS2 is yet unknown, but TMPRSS2-deficient mice lack a discernible phenotype suggesting functional redundancy (13). In 2006, we first identified TMPRSS2 as a virus-activating protease, by demonstrating that it cleaves the surface glycoprotein HA of human influenza A viruses (14). Subsequently, TMPRSS2 was shown to activate the fusion proteins of a number of other respiratory

[1]Institute of Virology, Philipps-University, Marburg, Germany   [2]Institute of Pharmaceutical Chemistry, Philipps-University, Marburg, Germany   [3]Department of Biomedical Sciences, Carlson College of Veterinary Medicine, Oregon State University, Corvallis, OR, USA   [4]Fraunhofer Institute for Molecular Biology and Applied Ecology, Gießen, Germany   [5]German Center for Infection Research (DZIF), Marburg-Gießen-Langen Site, Emerging Infections Unit, Philipps-University, Marburg, Germany

Correspondence: friebertshaeuser@staff.uni-marburg.de
*Dorothea Bestle, Miriam Ruth Heindl, and Hannah Limburg contributed equally to this work

viruses, including human metapneumovirus, human parainfluenza viruses, and CoVs, including SARS-CoV and Middle East respiratory syndrome (MERS)-CoV in vitro (reviewed in references 8 and 11). TMPRSS2 cleaves at single arginine or lysine residues (R/K↓), and hence, activates viral fusion proteins at the so called monobasic cleavage sites. More recent studies by us and others demonstrated that TMPRSS2-deficient mice do not suffer from pathology when infected with certain influenza A virus strains, SARS-CoV and MERS-CoV due to inhibition of proteolytic activation of progeny virus and consequently inhibition of virus spread along the respiratory tract (15, 16, 17, 18). These studies identified TMPRSS2 as an essential host cell factor for these respiratory viruses and further demonstrated that inhibition of virus activating host cell proteases, particularly TMPRSS2, provides a promising approach for the development of therapeutics to treat respiratory virus infections. The proprotein convertase furin is a type I transmembrane protein that is ubiquitously expressed in eukaryotic tissues and cells. Furin cleaves the precursors of a broad range of proteins, including hormones, growth factors, cell surface receptors, and adhesion molecules during their transport along the secretory pathway at multibasic motifs of the preferred consensus sequence R-X-R/K-R↓ (reviewed in reference 10). Moreover, furin has been identified as an activating protease for the fusion proteins of a broad range of viruses, including highly pathogenic avian influenza A viruses (HPAIV), HIV, Ebola virus, measles virus, and yellow fever virus as well as bacterial toxins such as Shiga toxin or anthrax toxin at multibasic motifs (reviewed in references 9, 10, and 19). Acquisition of a multibasic cleavage site by insertion of basic amino acids has long been known to be a prime determinant of avian influenza A virus pathogenicity in poultry. Activation of the surface glycoprotein HA of HPAIV by furin supports systemic spread of infection with often lethal outcome. In contrast, the HA of low pathogenic avian influenza A viruses (LPAIV) is activated at a monobasic cleavage site by trypsin-like serine proteases. Appropriate proteases are believed to be expressed only in the respiratory and intestinal tract of birds, confining spread of infection to these tissues.

Recent studies indicate that TMPRSS2 is also involved in SARS-CoV-2 S protein activation (20, 21). Transient expression of TMPRSS2 in Vero cells supports cathepsin-independent entry of SARS-CoV-2 pseudotypes. Moreover, pretreatment of human Caco-2 colon and human airway cells with the broad range inhibitor camostat mesylate, which can inhibit TMPRSS2 activity, markedly reduced the entry of SARS-CoV-2 as well as vesicular stomatitis virus pseudotypes containing the SARS-CoV-2 S protein. This suggests that a trypsin-like serine protease is crucial for SARS-CoV-2 entry into these cells. However, sequence analysis of the SARS-CoV-2 S protein suggests that furin may also be involved in S processing (Fig 1B; (22, 23)). The S1/S2 site of SARS-CoV-2 S protein contains an insertion of four amino acids providing a minimal furin cleavage site (R-R-A-$R_{685}$↓) in contrast to the S protein of SARS-CoV. Instead, similar to SARS-CoV the S2′ cleavage site of SARS-CoV-2, S possesses a paired dibasic motif with a single KR segment ($KR_{815}$↓) that is recognized by trypsin-like serine proteases.

In the present study, we demonstrate that the S protein of SARS-CoV-2 is activated by TMPRSS2 and furin. We also show that inhibitors against both proteases strongly suppress virus replication in human airway epithelial cells and that the combination of both

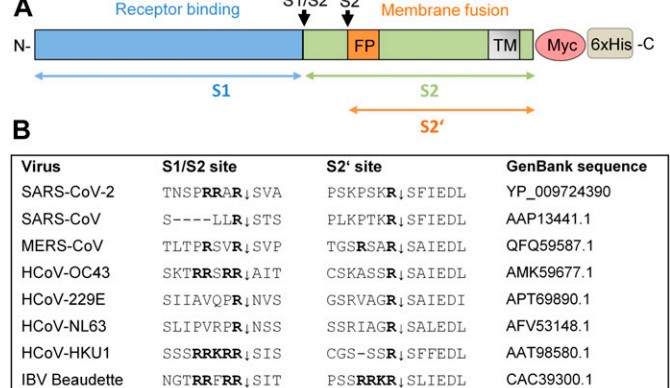

**Figure 1.  Cleavage of coronavirus S protein.**
**(A)** Schematic representation of the SARS-CoV-2 precursor and the S1 and S2 subunits. Fusion peptide (FP), and transmembrane domain (TM) are indicated. The S1/S2 and S2′ cleavage sites and subunits S1, S2, and S2′ are indicated by black and colored arrows, respectively. For immunochemical detection, recombinant S is expressed with a C-terminally fused Myc-6xHis-tag peptide in our study. **(B)** Alignment of the amino acid sequences at the S1/S2 and S2′ cleavage site of the S proteins of different human coronaviruses (HCoV) and avian infectious bronchitis virus strain Beaudette.

types of inhibitors produces a synergistic effect on virus reduction. Our results show that this approach has considerable therapeutic potential for treatment of COVID-19.

# Results

## Cleavage of SARS-CoV-2 S1/S2 site fluorescence resonance energy transfer (FRET)-substrates by furin

The S1/S2 cleavage site of the novel emerged SARS-CoV-2 has been shown to possess a minimal furin consensus motif of the sequence R-R-A-R↓ with an alanine instead of a basic residue in the P2 position (Fig 1B; (22, 23)). Only few furin substrates possess a nonbasic residue in the P2 position, such as *Pseudomonas aeruginosa* exotoxin A or Shiga toxin (10, 19). To test, whether the S1/S2 sequence of SARS-CoV-2 S protein is efficiently cleaved by furin, a small series of FRET substrates was synthesized (Fig 2A). All compounds possess a 3-nitrotyrosine amide as P4′ residue and a 2-amino-benzoyl fluorophore in the P7 position. The analogous sequences of the S proteins from MERS-CoV, SARS-CoV, and avian infectious bronchitis virus (IBV) strain Beaudette were prepared as reference substrates. Moreover, two FRET substrates of the SARS-CoV-2 S1/S2 cleavage site with P2 A → K and A → R mutations were synthesized, to evaluate whether they could constitute even more efficient cleavage sites for furin than the wild type. The FRET substrates were tested in an enzyme kinetic assay with human furin, and their cleavage efficiency is shown in Fig 2B. The FRET substrate of the SARS-CoV-2 S1/S2 cleavage site was efficiently cleaved by recombinant furin. In contrast, the monobasic SARS-CoV FRET substrate was not processed by furin. The MERS-CoV S1/S2 FRET substrate possessing a dibasic R-X-X-R motif was cleaved by furin ~10-fold less efficiently than the best substrates of this FRET

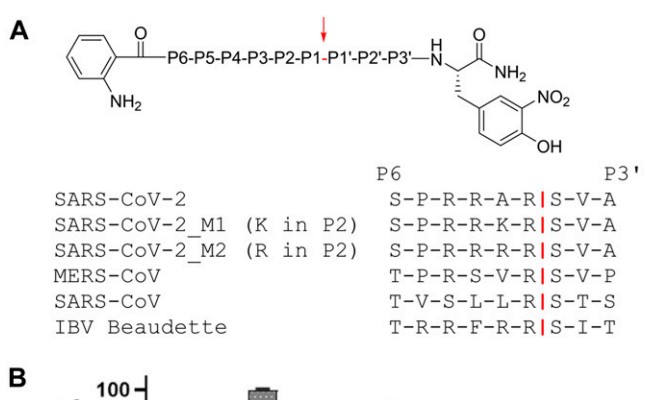

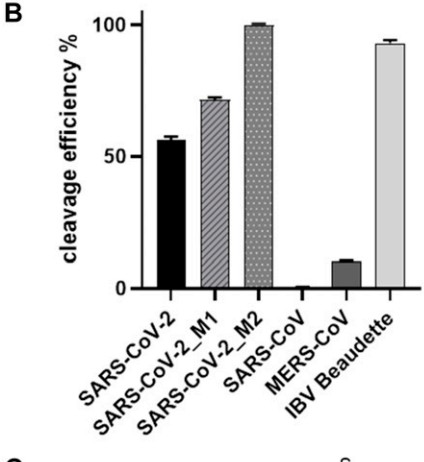

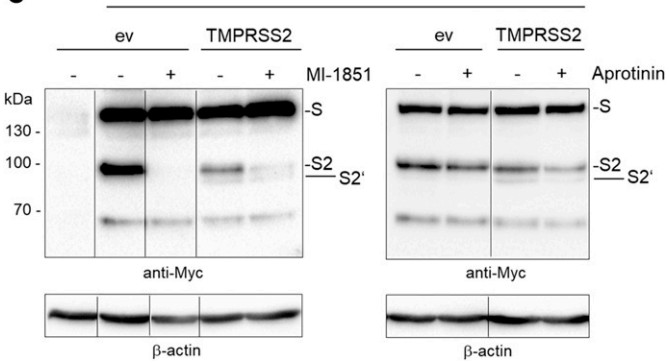

**Figure 2. Cleavage of SARS-CoV-2 S by furin and TMPRSS2.**
**(A)** Fluorescence resonance energy transfer substrates of the S protein S1/S2 sites of the indicated CoVs. M1 and M2 are mutants of the SARS-CoV-2 S1/S2 site with substitution of A → K or A → R in P2 position. IBV, avian infectious bronchitis virus strain Beaudette. Cleavage by furin is indicated in red. **(B)** Cleavage of the fluorescence resonance energy transfer substrates (20 $\mu$M) by furin (0.5 nM). Cleavage efficiency of SARS-CoV-2_M2 was set as 100%. **(C)** Cleavage of SARS-CoV-2 S by furin and TMPRSS2 in HEK293 cells. Cells were co-transfected with pCAGGS-S-Myc-6xHis and either empty vector or pCAGGS-TMPRSS2. Cells were then incubated in the absence or presence of aprotinin or furin inhibitor MI-1851 (50 $\mu$M each) for 48 h. Cell lysates were subjected to SDS–PAGE and Western blot analysis using antibodies against the C-terminal Myc-tag. For each Western blot lanes are spliced together from one immunoblot from one experiment. $\beta$-actin was used as loading control.
Source data are available for this figure.

series. The FRET substrate SARS-CoV-2_M1, which contains an optimized furin recognition site by virtue of an A → K mutation in the P2 position, was cleaved with similar efficiency compared with the wild-type sequence. However, substitution of A → R in the P2

position strongly enhanced cleavage by furin. As expected, the analogous reference sequence of IBV was also processed by furin very efficiently. The data show that the R-R-A-R motif at the S1/S2 cleavage site of SARS-CoV-2 S is efficiently cleaved by furin in vitro.

## SARS-Cov-2 spike protein is cleaved by both furin and TMPRSS2

We next examined whether the SARS-CoV-2 S protein is cleaved by endogenous furin in HEK293 cells. Cells were transiently transfected with pCAGGS plasmid encoding the SARS-CoV-2 S protein with a C-terminal Myc-6xHis-tag and incubated in the absence and presence of the potent synthetic furin inhibitor MI-1851 (cf. Fig S1; manuscript describing its synthesis submitted). At 48 h post transfection, cell lysates were subjected to SDS–PAGE and Western blot analysis using antibodies against the Myc epitope. As shown in Fig 2C (left panel), the uncleaved precursor S and the S2 subunit were detected in the absence of MI-1851, indicating that S is cleaved by endogenous proteases at the S1/S2 site in HEK293 cells. In contrast, S cleavage was efficiently prevented by MI-1851. The S1 subunit cannot be detected by the Myc-specific antibody (cf. Fig 1A). However, S cleavage was not prevented by the trypsin-like serine protease inhibitor aprotinin (Fig 2C, right panel, lane 2). Thus, the data indicate that SARS-CoV-2 S protein is cleaved by furin at the S1/S2 site in HEK293 cells.

We then investigated SARS-CoV-2 S cleavage by TMPRSS2. Because HEK293 cells do not express endogenous TMPRSS2 (unpublished data; see also www.proteinatlas.org), we co-transfected the cells with pCAGGS-S-Myc-6xHis and pCAGGS-TMPRSS2. Then, the cells were incubated in the absence or presence of MI-1851 to suppress S cleavage by endogenous furin. Interestingly, two S cleavage products of ~95 and 80 kD, respectively, were detected upon co-expression of TMPRSS2 in the absence of MI-1851 (Fig 2C, left panel), most likely S2 and S2′, as they can both be detected by the Myc-specific antibody (cf. Fig 1A). In the presence of MI-1851, only a minor S2 protein band was detected. However, the amount of S2′ protein present in transient TMPRSS2-expressing cells was similar in MI-1851–treated and untreated cells, suggesting that S cleavage at the S2′ site is only caused by TMPRSS2 activity. The small amount of S2 protein detected in TMPRSS2-expressing cells in the presence of MI-1851 was likely due to residual furin activity rather than cleavage of S at the S1/S2 site by TMPRSS2. Cleavage of S by TMPRSS2 at the S2′ site was further supported by the reduction of S2′ protein in aprotinin treated cells (Fig 2C, right panel). Together, the data show that SARS-CoV-2 S can be cleaved by furin and by TMPRSS2. The data further suggest that the proteases cleave S at different sites with furin processing the S1/S2 site and TMPRSS2 cleaving at the S2′ site.

## Knockdown of TMPRSS2 prevents proteolytic activation and multiplication of SARS-CoV-2 in Calu-3 human airway epithelial cells

Next, we investigated whether TMPRSS2 is involved in proteolytic activation and multicycle replication of SARS-CoV-2 in Calu-3 human airway epithelial cells. To specifically knockdown TMPRSS2 activity, we previously developed an antisense peptide-conjugated phosphorodiamidate morpholino oligomer (PPMO) (25). PPMOs are

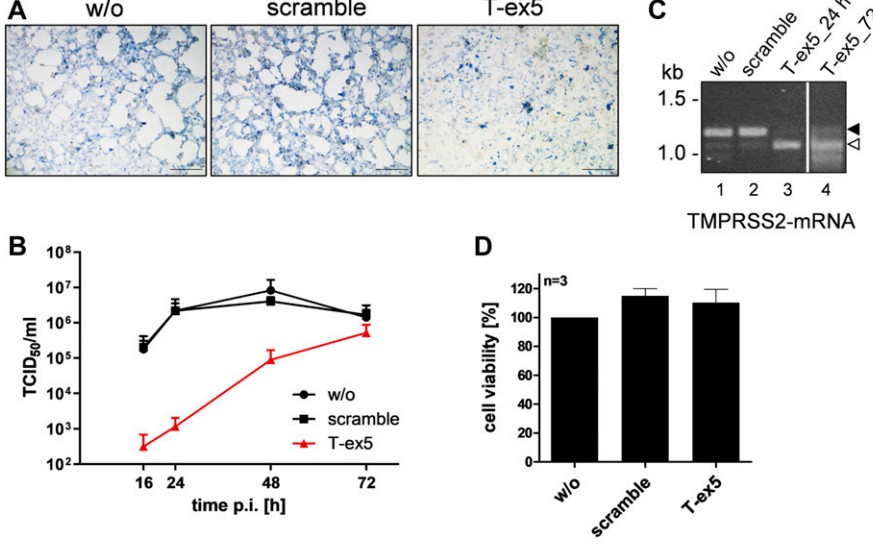

**Figure 3. Knockdown of TMPRSS2 expression by PPMO T-ex5 inhibits multicycle replication of SARS-CoV-2 in Calu-3 cells.**
**(A)** Multicycle replication of SARS-CoV-2 in T-ex5–treated Calu-3 cells. Cells were treated with 25 $\mu$M T-ex5 or control PPMO (scramble) for 24 h or remained without treatment (w/o). Cells were then inoculated with SARS-CoV-2 at a MOI of 0.001 for 1 h 30 min, the inoculum was removed and the cells further incubated in the absence of PPMO for 72 h. Cells were fixed and immunostained using a serum against SARS-CoV. Virus-positive cells are stained in blue. Scale bars indicate 500 $\mu$m. **(B)** Calu-3 cells were treated with PPMO for 24 h and then infected with SARS-CoV-2 for 72 h as described above. Virus titers in supernatants were determined by tissue culture infection dose 50% (TCID$_{50}$) end point dilution at indicated time points. Results are mean values ± SD of three independent experiments. **(C)** Analysis of TMPRSS2-mRNA in PPMO-treated Calu-3 cells. Cells were treated with 25 $\mu$M T-ex5, scramble PPMO or remained untreated (w/o) for 24 h (lanes 1–4). T-ex5–treated cells were inoculated with SARS-CoV-2 as described above and incubated in the absence of PPMO for 72 h (lane 4). Total RNA was isolated and analyzed by RT-PCR using primers designed to amplify 1,228 nt of full-length TMPRSS2-mRNA. Full-length and truncated PCR products lacking exon 5 are indicated by filled and open arrow heads, respectively. **(D)** Effect of PPMO treatment on Calu-3 cell viability. Calu-3 cells were treated with scramble or T-ex5 PPMO (25 $\mu$M) for 24 h. Cell viability of untreated (w/o) cells was set as 100%. Results are mean values ± SD (n = 3).
Source data are available for this figure.

single-stranded nucleic acid–like compounds, composed of a morpholino oligomer covalently conjugated to a cell-penetrating peptide, and can interfere with gene expression by sterically blocking complementary RNAs. PPMOs are water-soluble and achieve entry into cells and tissues without assisted delivery (reviewed in references 26 and 27). The previously developed PPMO T-ex5 interferes with splicing of TMPRSS2 pre-mRNA, resulting in the production of mature mRNA lacking exon 5 and consequently expression of a truncated TMPRSS2 form that is enzymatically inactive. Using T-ex5 PPMO-mediated knockdown of TMPRSS2 activity, we were able to identify TMPRSS2 as the major influenza A virus activating protease in Calu-3 cells and primary human airway epithelial cells and of influenza B virus in primary human type II pneumocytes (25, 28).

Here, Calu-3 cells were treated once with T-ex5 PPMO for 24 h before infection with SARS-CoV-2 to inhibit the production of normal TMPRSS2-mRNA and deplete enzymatically active TMPRSS2 present in the cells. The cells were then inoculated with SARS-CoV-2 at a low MOI of 0.001, further incubated without additional PPMO treatment for 72 h, and then fixed and immunostained using a rabbit serum originally produced against 2002 SARS-CoV. As shown in Fig 3A, a strong cytopathic effect (CPE) and efficient spread of SARS-CoV-2 infection was visible in Calu-3 cells treated with a negative-control PPMO of nonsense sequence designated as "scramble" as well as untreated cells that were used as controls. In contrast, no CPE and only small foci of infection were observed in T-ex5 PPMO-treated cells at 72 h p.i. (Fig 3A). To examine SARS-CoV-2 activation and multicycle replication in PPMO-treated cells in more detail, Calu-3 cells were treated with PPMO for 24 h before infection, then inoculated with virus at an MOI of 0.001 for 1 h 30 min, and incubated for 72 h in the absence of further PPMO, as described above. At different time points, virus titers in supernatants were

determined by tissue culture infection dose 50% (TCID$_{50}$) end point dilution. T-ex5 PPMO treatment dramatically reduced virus titers in Calu-3 cells, by 500- and 2,000-fold at 16 and 24 h p.i., respectively, and 90-fold at 48 h p.i. (Fig 3B).

To confirm knockdown of enzymatically active TMPRSS2 expression, Calu-3 cells were treated with PPMO or remained untreated for 24 h, after which TMPRSS2-specific mRNA was isolated and analyzed by RT-PCR as described previously (25). Total RNA was analyzed with primers designed to amplify nucleotides 108–1,336 of TMPRSS2-mRNA. A full-length PCR product of 1,228 bp was amplified from untreated and scramble PPMO-treated Calu-3 cells, whereas a shorter PCR fragment of about 1,100 bp was amplified from T-ex5 PPMO-treated cells (Fig 3C). Sequencing revealed that the truncated TMPRSS2-mRNA lacked the entire exon 5 (data not shown). To further confirm that T-ex5 PPMO single dose treatment before infection still interferes with TMPRSS2-mRNA splicing at 72 h p.i., total RNA was isolated from infected cells at 72 h p.i. and amplified as described above. As shown in Fig 3C, most TMPRSS2-mRNA amplified from T-ex5-treated cells at 72 h p.i. lacked exon 5. The data demonstrate that T-ex5 was very effective at producing exon skipping in TMPRSS2-pre-mRNA and, thus, at inhibiting expression of enzymatically active protease, during the virus growth period in Calu-3 cells. However, a small band of the full-length PCR product was visible after 72 h p.i., indicating low levels of expression of enzymatically active TMPRSS2 at later time points of the virus growth period, which may explain the increase in virus titers observed at 48 h p.i. (cf. Fig 3B). Cell viability was not affected by T-ex5 PPMO treatment of Calu-3 cells, as shown in Fig 3D and described previously (25, 28).

Together, our data identify TMPRSS2 as a host cell factor essential for SARS-CoV-2 activation and multiplication in Calu-3 cells and show that down-regulation of TMPRSS2 activity dramatically blocks SARS-CoV-2 replication.

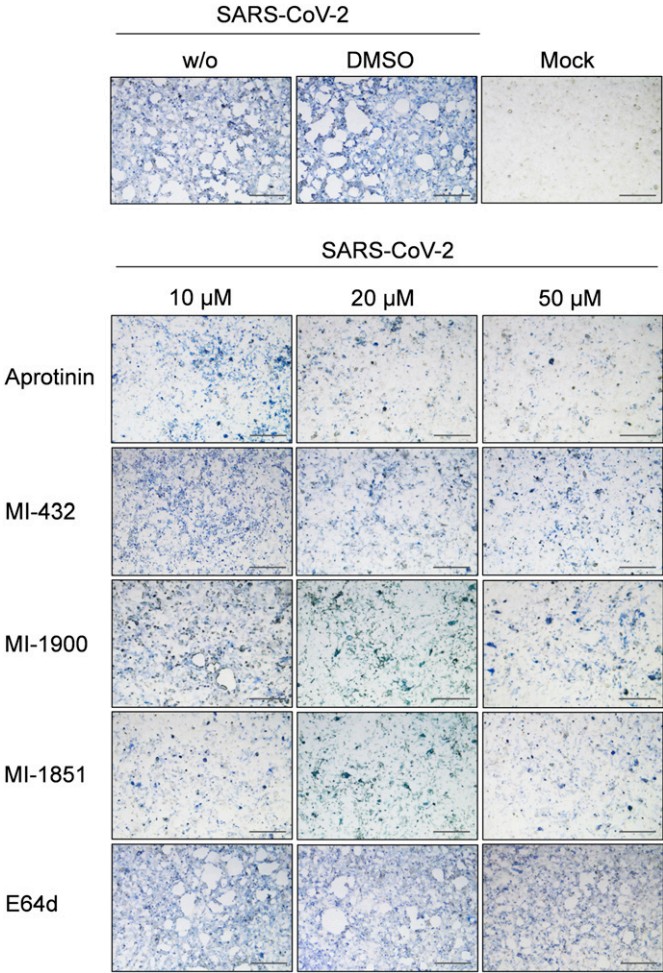

**Figure 4. Inhibition of SARS-CoV-2 multiplication in human airway cells by inhibitors of furin and TMPRSS2.**
Calu-3 cells were inoculated with SARS-CoV-2 at a low MOI of 0.001 and then incubated in the presence of inhibitors of TMPRSS2 (aprotinin, MI-432, and MI-1900), furin (MI-1851), and endosomal cathepsins (E64d), respectively, at the indicated concentrations. Cells were fixed and immunostained using a rabbit serum against SARS-CoV at 72 h p.i. Virus-positive cells are stained in blue or dark gray depending on the staining intensity. Cells infected in the absence of inhibitors (w/o), in the presence of DMSO (0.5%) and noninfected cells (mock) were used as controls. Scale bars indicate 500 μm. Images are representatives of three independent experiments.

## Inhibition of either TMPRSS2 or furin activity suppresses multicycle replication of SARS-CoV-2 in human airway epithelial cells

We next investigated the efficacy of different inhibitors of trypsin-like serine proteases, also inhibiting TMPRSS2, on preventing SARS-CoV-2 activation by TMPRSS2 in Calu-3 cells. We used the natural broad-range serine protease inhibitor aprotinin from bovine lung and two prospective peptide mimetic inhibitors of TMPRSS2, MI-432 (29), and MI-1900 (Fig S1). Aprotinin has long been known to prevent proteolytic activation and multiplication of influenza A virus in cell cultures and mice. Furthermore, inhalation of aerosolized aprotinin by influenza patients markedly reduced the duration of symptoms without causing side effects (30). MI-432 was shown to efficiently

inhibit proteolytic activation and multiplication of influenza A virus in Calu-3 cells (31). The inhibitor MI-1900 is a monobasic and structurally related analog of the dibasic inhibitor MI-432.

To examine the antiviral efficacy of the protease inhibitors against SARS-CoV-2, Calu-3 cells were infected with the virus at a low MOI of 0.001 for 1 h 30 min, after which the inoculum was removed and the cells incubated in the presence of the inhibitors at the indicated concentrations for 72 h. The cells were fixed and immunostained using an antiserum against 2002 SARS-CoV. As shown in Fig 4, strong CPE, with holes visible throughout the monolayer, and efficient spread of SARS-CoV-2 infection was observed in Calu-3 cells in the absence of protease inhibitors. Spread of SARS-CoV-2 infection and virus-induced CPE was efficiently inhibited by aprotinin treatment in a dose-dependent manner and only a few virus-positive cells were visible in Calu-3 cultures treated with 20 and 50 μM aprotinin. Even at a lower concentration of 10 μM, the spread of SARS-CoV-2 was greatly reduced and CPE markedly prevented. Treatment with peptide mimetic TMPRSS2 inhibitors MI-432 and MI-1900 also strongly prevented SARS-CoV-2 multiplication and CPE in Calu-3 cells in a dose-dependent manner, although less potently than aprotinin. At 20 or 50 μM of MI-432 or MI-1900, only small foci of infection were visible. At a concentration of 10 μM, virus spread and CPE in MI-432– or MI-1900–treated cells were still reduced compared with control cells. The data demonstrate that SARS-CoV-2 multiplication in Calu-3 human airway cells can be strongly suppressed by aprotinin and the synthetic TMPRSS2 inhibitors MI-432 and MI-1900.

The observed efficient cleavage of transient expressed SARS-CoV-2 S protein by furin in HEK293 cells prompted us to investigate if furin is involved in SARS-CoV-2 activation in Calu-3 cells. Hence, virus spread in Calu-3 cells was analyzed in the presence of the furin inhibitor MI-1851. Interestingly, MI-1851 strongly inhibited SARS-CoV-2 spread at even the lowest concentration of 10 μM, indicating that furin is critical for SARS-CoV-2 activation and multiplication in these cells (Fig 4). Finally, to examine whether endosomal cathepsins are involved in SARS-CoV-2 activation in Calu-3 cells, multicycle virus replication was determined in the presence of the cathepsin inhibitor E64d. Cathepsin L was shown to cleave the S protein of 2002 SARS-CoV S close to the S1/S2 site (R667) at T678 in vitro (2). Here, strong CPE and foci of infection were observed in E64d-treated cells even at the highest dose of 50 μM, similar to that observed in DMSO-treated as well as untreated control cells, indicating that SARS-CoV-2 activation in Calu-3 cells is independent of endosomal cathepsins.

In sum, our data demonstrate that inhibition of either TMPRSS2 or furin strongly inhibits SARS-CoV-2 in Calu-3 human airway cells, indicating that both proteases are critical for S activation. In contrast, endosomal cathepsins are dispensable or not involved at all in SARS-CoV-2 activation in these cells.

## Growth kinetics of SARS-CoV-2 in protease inhibitor–treated Calu-3 cells

To analyze inhibition of SARS-CoV-2 activation and multiplication by the different protease inhibitors in more detail, we performed virus growth kinetics in inhibitor-treated cells. Calu-3 cells were inoculated with SARS-CoV-2 at an MOI of 0.001 and then incubated

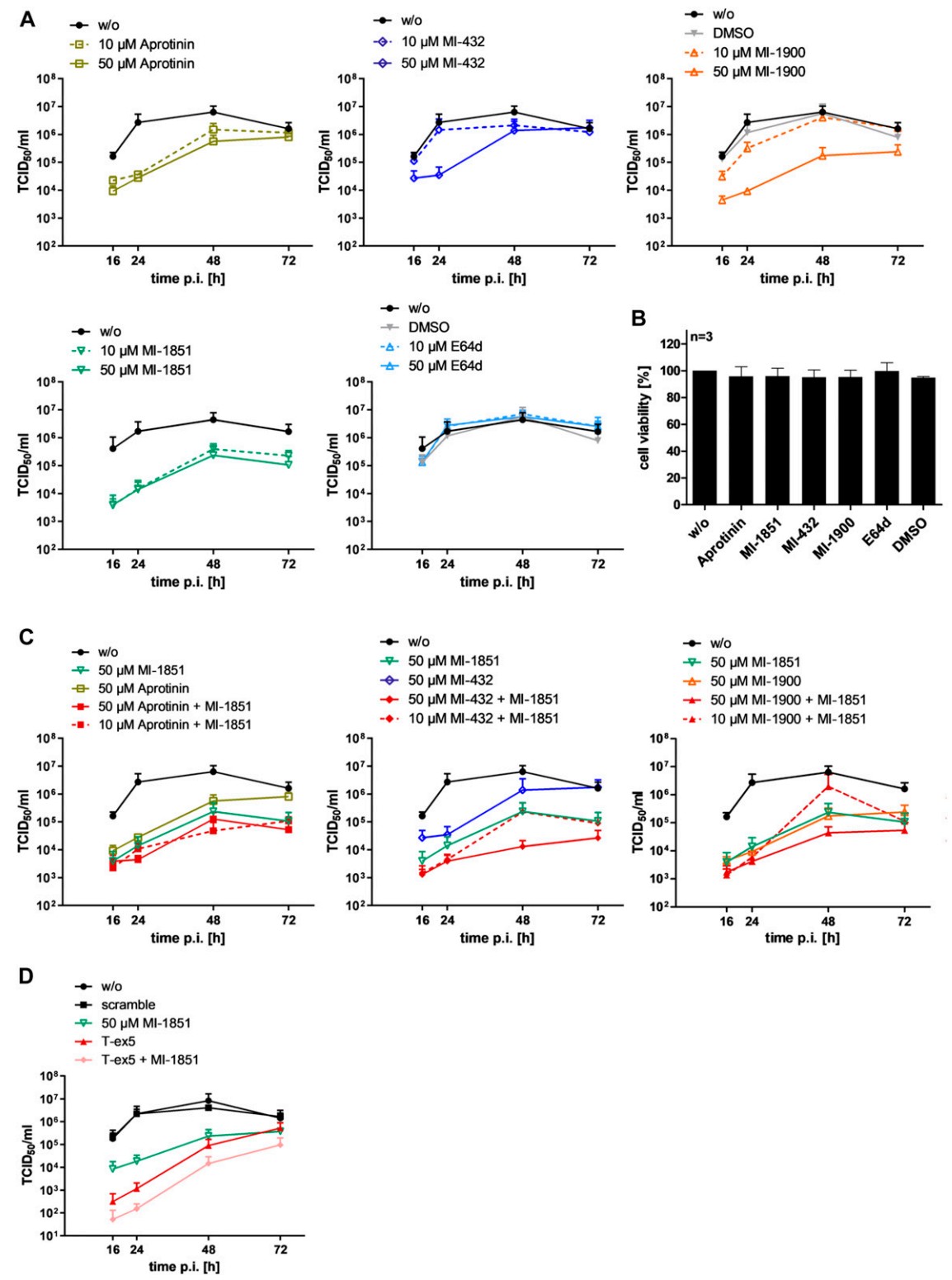

**Figure 5. Inhibition of SARS-CoV-2 multicycle replication in human airway epithelial cells by inhibitors of TMPRSS2 and furin.**
**(A)** Calu-3 cells were inoculated with SARS-CoV-2 at a low MOI of 0.001 and then incubated in the absence (w/o) or presence of inhibitors of TMPRSS2 (aprotinin, MI-432, and MI-1900), furin (MI-1851), and endosomal cathepsins (E64d), respectively, or DMSO (0.5%), at the indicated concentrations. At 16, 24, 48, and 72 h postinfection (p.i.), supernatants were collected, and virus replication was determined by tissue culture infection dose 50% ($TCID_{50}$) titration at indicated time points. Data are mean values ± SD of three to five independent experiments. **(B)** Effect of inhibitor treatment on cell viability. Calu-3 cells were treated with the indicated protease inhibitor (50 $\mu$M) for 72 h. Untreated cells (w/o) and DMSO treated cells were used as controls. Cell viability of untreated cells was set as 100%. Results are mean values ± SD (n = 3). **(C)** Antiviral

in the presence of 10 or 50 $\mu M$ of the different protease inhibitors. At 16, 24, 48, and 72 h p.i., the viral titer in supernatants was determined by TCID$_{50}$ titration. Untreated cells and cells treated with DMSO alone were used as controls. SARS-CoV-2 replicated to high titers within 24–48 h in untreated and DMSO-treated cells Calu-3 cells (Fig 5A). Aprotinin suppressed virus replication 25- to 100-fold compared with control cells even at a concentration of 10 $\mu M$ at 16-48 h p.i. The TMPRSS2 inhibitor MI-432 only slightly affected virus replication at a concentration of 10 $\mu M$ but reduced virus titers 75-fold at 24 h p.i. at 50 $\mu M$. Treatment of cells with TMPRSS2 inhibitor MI-1900 reduced virus titers in a dose-dependent manner and caused strong inhibition of SARS-CoV-2 replication at 50 $\mu M$ with 35- to 280-fold reduced viral titers compared with control cells. The furin inhibitor MI-1851 efficiently suppressed SARS-CoV-2 multiplication in Calu-3 cells, producing a 30- to 190-fold reduction in virus titers at a dose of 10 $\mu M$. In contrast, virus multiplication was not affected by treatment with the cathepsin inhibitor E64d, consistent with the data shown in Fig 4. To provide evidence that inhibition of SARS-CoV-2 replication in inhibitor-treated cells was not caused by cytotoxic effects, we analyzed cell viability in Calu-3 cells treated with 50 $\mu M$ of the different inhibitors for 72 h. As shown in Fig 5B, evaluation of cell viability revealed no significant cytotoxicity by any of the inhibitors under conditions used in the virus growth experiments.

The data demonstrate that SARS-CoV-2 replication can be efficiently reduced by inhibiting either TMPRSS2 or furin activity, demonstrating that both proteases are crucial for SARS-CoV-2 activation.

### Treatment of SARS-CoV-2–infected Calu-3 cells with a combination of TMPRSS2 and furin inhibitors

Finally, we wished to examine whether the combination of inhibitors against TMPRSS2 and furin shows a synergistic antiviral effect. Therefore, Calu-3 cells were infected with virus as described above and incubated in the presence of aprotinin, MI-432, or MI-1900 in combination with MI-1851 at 10 and 50 $\mu M$ each, respectively, for 72 h. Virus titers in supernatants were determined at indicated time points. Single-dose treatment of each inhibitor and untreated cells were used as controls. As shown in Fig 5C, the combination of 10 $\mu M$ of MI-1851 with either aprotinin or MI-432 showed enhanced antiviral activity against SARS-CoV-2 and 10- to 30-fold reduced virus titers compared with 10 $\mu M$ of each inhibitor alone and also reduced viral titer four to eightfold more than that observed with either of the single inhibitors at 50 $\mu M$. A combination of 50 $\mu M$ each of MI-432 and MI-1851 reduced virus titers 10- to 32-fold compared with 50 $\mu M$ of each inhibitor alone and thereby dramatically suppressed SARS-CoV-2 multiplication 100- to 250-fold compared with untreated or DMSO-treated cells. In contrast, treatment of Calu-3 cells with 50 $\mu M$ each of MI-1851 and aprotinin did not cause

further suppression of virus titers compared with the combination of 10 $\mu M$ of each inhibitor. The combination of 10 $\mu M$ each of MI-1851 and MI-1900 did not show enhanced antiviral activity compared with single inhibitor treatments at 10 $\mu M$. However, treatment of cells with 50 $\mu M$ each of MI-1900 and MI-1851 caused fivefold reduction in viral titers when compared with cells treated with 50 $\mu M$ of each inhibitor alone and thereby SARS-CoV-2 multiplication in Calu-3 cells was markedly reduced compared to control cells. We furthermore examined the antiviral activity of a combination of T-ex5 PPMO and furin inhibitor MI-1851 against SARS-CoV-2 in Calu-3 cells. As shown in Fig 5D, combined treatment of Calu-3 cells with 25 $\mu M$ T-ex5 PPMO and 50 $\mu M$ MI-1851 almost completely blocked SARS-CoV-2 replication with a nearly 15,000-fold reduction in virus titers at 24 h p.i., and reduced virus titers 500-fold at 48 h p.i. compared with control cells. Combination of T-ex5 and MI-1851 was synergistic and caused 30- to 10-fold lower virus titers at 16 and 24 h p.i. compared with single inhibitor-treated cells. The data demonstrate that efficient inhibition of S cleavage by a combination of TMPRSS2 and furin inhibitors can dramatically block SARS-CoV-2 replication in human airway epithelial cells. Furthermore, our data show that combination of TMPRSS2 and furin inhibitors can act synergistically to produce inhibition of SARS-CoV-2 activation and multiplication at lower doses than single protease inhibitor treatment.

In conclusion, our data demonstrate that both TMPRSS2 and furin cleave the SARS-CoV-2 S protein and are essential for efficient virus multicycle replication in Calu-3 human airway cells. The results indicate that TMPRSS2 and furin cleave S at different sites—furin at the S1/S2 site and TMPRSS2 at the S2′ site—and suggest that TMPRSS2 and furin cannot compensate for each other in SARS-CoV-2 S activation. Our data further demonstrate that inhibition of either one of these critical proteases can render the S protein of SARS-CoV-2 unable to efficiently mediate virus entry and membrane fusion and, therefore, provides a promising therapeutic approach for treatment of COVID-19.

## Discussion

Proteolytic processing of CoV S is a complex process that requires cleavage at two different sites and is yet not fully understood. The amino acid sequence at the S1/S2 and S2′ cleavage sites varies among CoVs (Fig 1B), suggesting that differing proteases may be involved in activation of different CoVs. Sequence analyses of the S protein of the emerged SARS-CoV-2 suggested that the R-R-A-R motif at the S1/S2 site may be sensitive to cleavage by furin, whereas the S2′ site contains a single R residue that can be cleaved by trypsin-like serine proteases such as TMPRSS2 (20, 22, 23). In the present study, we demonstrate that the SARS-CoV-2 S protein is cleaved by furin and by TMPRSS2. Furthermore, we show that

activity of combinations of TMPRSS2 and furin inhibitors against SARS-CoV-2 in human airway epithelial cells. Calu-3 cells were inoculated with SARS-CoV-2 at an MOI of 0.001 as described above and then incubated in the presence of single protease inhibitors or inhibitor combinations at the indicated concentrations. Virus titers in supernatants were determined by TCID$_{50}$ at 16, 24, 48, and 72 h p.i. Data are mean values ± SD of three independent experiments. **(D)** Calu-3 cells were treated with PPMO for 24 h, then infected with SARS-CoV-2 as described above and incubated in the absence of PPMO (w/o, scramble and T-ex5) and with or without 50 $\mu M$ of furin inhibitor treatment (MI-1851) for 72 h. At 16, 24, 48, and 72 h p.i., supernatants were collected, and viral titers were determined by TCID$_{50}$ at indicated time points. Data are mean values ± SD (n = 2).

multicycle replication of SARS-CoV-2 in Calu-3 human airway cells is strongly suppressed by inhibiting TMPRSS2 and furin activity, demonstrating that both proteases are crucial for S activation in these cells. Our data indicate that furin cleaves at the S1/S2 site, whereas TMPRSS2 cleaves at the S2′ site. The effective processing of the S1/S2 site by furin was additionally confirmed by comparing the cleavage rates of various FRET substrates derived from the P6-P3′ segments of SARS-CoV-2 and other CoVs. The data clearly revealed that due to the 4-mer PRRA insertion, a well-suited furin cleavage site exists in the S of SARS-CoV-2, which is similarly cleaved as the sequence from the IBV CoV, whereas the analogous substrate of SARS-CoV is not processed by furin. Strong inhibition of SARS-CoV-2 replication in Calu-3 cells by synthetic furin inhibitor MI-1851 furthermore suggests that TMPRSS2 does not compensate for furin cleavage at the S1/S2 site. Likewise, strong inhibition of SARS-CoV-2 replication by knockdown of TMPRSS2 activity using T-ex5 PPMO or treatment of Calu-3 cells with aprotinin, MI-432 and MI-1900, respectively, indicates that furin cannot compensate for the lack of TMPRSS2 in S activation. This was further confirmed by using an analogous FRET substrate derived from the S2′ cleavage site of the SARS-CoV-2 S protein (Fig S2). Kinetic measurements clearly revealed that this substrate cannot be cleaved by furin (Fig S2). Thus, we could experimentally demonstrate for the first time that furin only activates the S1/S2 site, as expected from the amino acid sequence at the cleavage sites (22, 23). Together, our data indicate that furin and TMPRSS2 cleave S at different sites and that cleavage by both proteases is crucial to render the S protein active for mediating virus entry and membrane fusion (Fig 6). Iwata-Yoshikawa et al (2019) (18) showed that TMPRSS2-deficient mice do not develop disease symptoms when infected with SARS-CoV and MERS-CoV (18). The data demonstrated that TMPRSS2 is essential for multicycle replication and spread of these CoVs in a manner similar to what we and others have observed for certain influenza A virus strains (15, 16, 17). However, it remains to be determined whether knockout of TMPRSS2 prevents cleavage of the S proteins of SARS-CoV and MERS-CoV at both sites, S1/S2 and S2′, or whether another protease is involved in S cleavage, similar to what we have observed here for SARS-CoV-2.

In cell culture, CoVs can enter cells via two distinct routes: the late endosome where S is cleaved by cathepsins or via the cell surface or early endosome using trypsin-like proteases for S cleavage ((2, 32) reviewed in reference 7). However, several recent studies revealed that clinical isolates of human CoV (HCoV) achieve activation by trypsin-like serine proteases and use endosomal cathepsins only in the absence of suitable trypsin-like proteases in cell culture (33, 34). Thus, activation by cathepsins appears to be a mechanism that is acquired by the virus during multiple passaging in cell cultures (34). Congruently, Zhou et al (2015) (35) showed that SARS-CoV pathogenesis in mice was strongly prevented by camostat, a broad-range inhibitor of trypsin-like serine proteases, but not by inhibitors of endosomal cathepsins (35). Here, we show that the cysteine protease inhibitor E64d, which also inhibits cathepsin L and B, did not affect SARS-CoV-2 replication in Calu-3 cells, indicating that endosomal cathepsins are dispensable or not involved at all in SARS-CoV-2 activation in human airway cells.

The presence of a multibasic cleavage site that is processed by ubiquitously expressed furin and, therefore, supports systemic

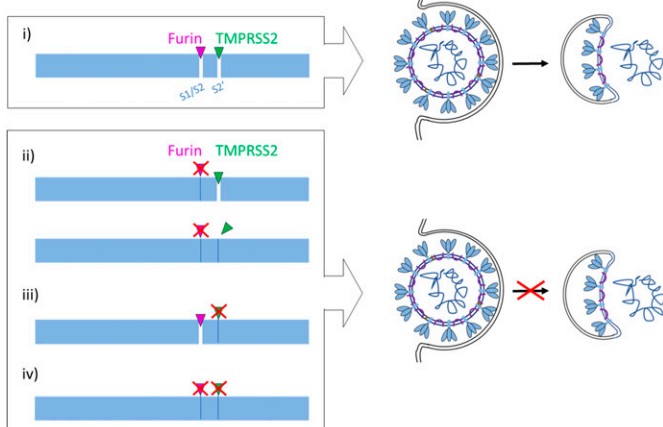

**Figure 6.  Proposed processing of SARS-CoV-2 spike protein S by TMPRSS2 and furin.**
(i) S must be cleaved at two sites, S1/S2 and S2′, to trigger fusion of viral and cellular membranes during virus entry to release the virus genome into the host cell. CoV S cleavage is believed to occur sequentially, with cleavage at the S1/S2 site occurring first and subsequent cleavage at the S2′ site. Furin processes the S1/S2 site, whereas TMPRSS2 cleaves at the S2′ site, and both proteases cannot compensate each other. Inhibition of either furin (ii) or TMPRSS2 (iii) or simultaneous inhibition of both proteases (iv) renders the S protein fusion-inactive and prevents virus entry. Inhibition of TMPRSS2 prevents exposure of the fusion peptide at the N-terminus of the S2′ subunit (iii, iv). Inhibition of furin cleavage at the S1/S2 site may directly interfere with virus entry and membrane fusion by steric blockage of conformational changes (ii, upper scheme) or may prevent exposure of the S2′ site to TMPRSS2 (ii, lower scheme).

spread of infection, has long been known to be an important determinant of the pathogenicity of HPAIV as well as virulent strains of Newcastle disease virus in poultry (reviewed in references 11 and 36). The S protein of IBV strain Beaudette contains multibasic motifs at the S1/S2 and S2′ site (Fig 1B). IBV belongs to the genus *Gammacoronavirus* and causes a highly contagious, acute respiratory disease of chickens. Cleavage of IBV S protein by furin at the S2′ site has been associated with neurotropism in chicken (37). Congruently, here, FRET substrates of the S1/S2 and S2′ site of the IBV Beaudette S protein were efficiently cleaved by furin (Figs 2B and S2B). However, the contribution of furin-cleavable multibasic motifs at the S1/S2 and/or S2′ site with regard to multicycle replication, cellular tropism, and pathogenicity of HCoVs remains to be determined. HCoV-OC43 and HCoV-HKU1 possess a furin cleavage motif at the S1/S2 site. In contrast, the S proteins of the 2002 SARS-CoV, HCoV-229E, and HCoV-NL63 possess single arginine residues at both cleavage sites (see also Figs 2B and S2A). Interestingly, among the S proteins of the seven CoV infecting humans, only SARS-CoV S lacks the 4-mer insertion at the S1/S2 site (Fig 1B; (23)). The S protein of MERS-CoV contains a dibasic motif of the sequence R-X-X-R at both S1/S2 and S2′ sites. However, it is still controversial whether MERS-CoV is activated by furin in human airway epithelial cells (38, 39, 40, 41). Other proteases such as the serine protease matriptase/ST14, which also prefers sequences with arginine in the P1 and P4 position, might be involved. Matriptase is expressed in a broad range of cells and tissues and has been shown to activate the HA of H9N2 influenza A viruses possessing the cleavage site motif R-S-S-R ((42), reviewed in reference 11). Interestingly, a study by Park et al. indicated that cleavage of MERS-CoV S by furin or other proprotein

convertases at the S1/S2 site takes place in virus-producing cells before virus release and can impact the cellular localization of membrane fusion and virus entry into a new cell (43). Cleavage of MERS-CoV at the S1/S2 site was postulated as a prerequiste for subsequent cleavage of S at the S2′ site by host proteases present at the surface or in early endosomes of human airway cells, thus facilitating virus entry independent of S cleavage by cathepsins in the late endosome. However, other HCoVs, including the 2002 SARS-CoV, are reportedly released with non-cleaved S from infected cells. Hence, S cleavage at both sites, S1/S2 and S2′, has to take place at the stage of entry of these viruses.

The present study demonstrates that TMPRSS2 and furin are promising drug targets for the treatment of COVID-19. Furin and/or TMPRSS2 inhibitors could be used alone or in combination to target either or both of these proteases. The TMPRSS2 inhibitors MI-432 and MI-1900 as well as the furin inhibitor MI-1851 provide promising compounds for further drug development. MI-1900 and MI-1851 markedly reduced virus titers even at 72 h p.i., although the cells were treated only once after the inoculum was removed. In aprotinin- or MI-432–treated cells, an increase in viral titers was observed at 48 h p.i., and at 72 h p.i., virus titers were similar to control cells. An additional treatment of cells with protease in-hibitors at 24 h p.i. may well have supported efficient blockage of virus inhibition and multiplication at later time points. This needs to be investigated in more detail in future studies. In search for suitable antiviral therapies against SARS-CoV-2 infections, protease inhibitors that have been approved for other applications may be promising for drug repurposing to treat COVID-19. Aprotinin is a broad range serine protease inhibitor isolated from bovine lung, used as a fibrinolysis inhibitor to reduce perioperative bleeding (reviewed in reference 44) and has long been known to inhibit influenza A virus activation and replication in cell culture and in mice in vivo (30). In a clinical trial, inhalation of aerosolized aprotinin in patients with influenza and parainfluenza markedly reduced the duration of symptoms without causing side effects (30). Thus, aprotinin is an inhibitor of TMPRSS2 worthy of consideration for further testing and possible development as a therapeutic treatment for COVID-19. Another promising TMPRSS2 inhibitor candidate for COVID-19 treatment is the broad range protease inhibitor camostat mesylate, which is approved for the treatment of pancreatitis (45, 46). Camostate mesylate has been shown to efficiently inhibit replication of different CoV in cell culture and experimentally infected mice (35, 47, 48). Recently, Hoffmann et al (2020) (20) showed that pretreatment of human Caco-2 colon cells and human airway cells with camostat mesylate markedly reduced entry of SARS-CoV-2 as well as vesicular stomatitis virus pseudotypes containing the SARS-CoV-2 S protein (20).

However, it should be noted that all of these compounds inhibit numerous trypsin-like serine proteases and thus may cause various adverse effects. A specific inhibition of TMPRSS2 activity during an acute SARS-CoV-2 infection would provide the most promising approach to reduce side effects by inhibiting virus activation by host cell proteases. TMPRSS2-deficient mice show no discernible phenotype, indicating functional redundancy or compensation of physiological functions by other protease(s) in the host (13). Un-fortunately, there is no crystal structure of TMPRSS2 available so far, which prohibits a rational structure-based design of more efficient inhibitors of this protease. However, first homology models have

been established, which may help for the development of improved TMPRSS2 inhibitors in the future (49, 50 Preprint).

PPMO are highly selective inhibitors of target gene expression. They bind to a complementary sequence in target mRNA and can affect gene expression by steric blockage of translation initiation or pre-mRNA splicing. The demonstration of T-ex5 PPMO efficacy in the present study suggests that reducing TMPRSS2 expression by use of an mRNA-directed approach in general and by PPMO in particular is worthy of further consideration. Importantly, in various experimental animal models of other respiratory virus infections and disease, PPMO were able to be transported to lung tissue after intranasal administration and produced strong reductions in virus growth and virus-induced pathology (51, 52, 53, 54).

Very effective furin inhibitors containing a C-terminal 4-amidi-nobenzylamide residue have been developed in recent years. Several of these analogs have been successfully used to inhibit the replication of numerous furin dependent human pathogenic vi-ruses such as H5N1 influenza A virus, Chikungunya virus, West Nile virus and dengue-2 virus, mumps virus or respiratory syncytial virus (reviewed in references 49, 55, and 56). So far, these inhibitors have been used only in virus-infected cell cultures and not in animal models. However, the less potent furin inhibitor hexa-D-arginine has been used in mice and rats and protected them against P. aeruginosa exotoxin A and anthrax toxemia (57, 58). Therefore, it can be speculated that a specific furin inhibition in the respiratory tract and lungs by inhalative treatment of, for example, MI-1851 or structurally related compounds could be possible without severe side reactions, despite the many physiological functions of furin.

Here, the combination of the TMPRSS2 inhibitors aprotinin or MI-432 with furin inhibitor MI-1851 showed enhanced antiviral activity against SARS-CoV-2 in human airway cells and supported strong reduction of virus multiplication at lower doses compared with treatment with each inhibitor alone. Therefore, the combination of TMPRSS2 and furin inhibitors provides a promising therapeutic strategy for treatment of SARS-CoV-2 infections that not only may enhance antiviral effects but may also reduce drug toxicity and undesirable side effects by allowing reductions of the inhibitor doses. Notably, inhibition of TMPRSS2 and furin acts on the same target and our data show that inhibition of S cleavage at one of the two sites is sufficient to suppress SARS-CoV-2 replication by reducing the pro-duction of infectious progeny virus containing inactive S. Thus, the combination of TMPRSS2 and furin inhibitors can act synergistically until S cleavage at one or two sites is prevented. The combination of protease inhibitors with antiviral compounds provides an approach that may produce yet more synergistic antiviral activity at lower drug doses and may furthermore exclude the development of drug-resistant viruses. The combination of TMPRSS2 and furin inhibitors, respectively, with the neuraminidase inhibitor oseltamivir carboxylate has been shown to block influenza A virus replication in human airway cells at remarkably lower concentration of each inhibitor as compared with single inhibitor treatment (59). Combination of a furin inhibitor with oseltamivir carboxylate and the antiviral compounds ribavirin and favipiravir, respectively, efficiently blocked multicycle replication of HPAIV of subtype H5N1 and H7N1 in cell cultures (60, 61). Thus, combination of protease inhibitors (e.g., aprotinin or camostat) and antivirals provides a promising strategy for the treatment of COVID-19 and should be tested in cell cultures and animal models.

In summary, we demonstrate that TMPRSS2 and furin are essential for activation and multiplication of the novel emerged SARS-CoV-2 in human airway epithelial cells and provide promising drug targets for treatment of COVID-19. TMPRSS2 and furin have been shown to be involved in the proteolytic activation of a broad range of viruses. However, the development of host protease inhibitors as a preventative and/or therapeutic strategy for the treatment of virus infections has been minimal to date. Our data demonstrate the high potential of protease inhibitors as drugs for SARS-CoV-2 treatment and highlight the rationale of drug development and repurposing of host protease inhibitors for the treatment of virus infections in general and emerging CoV infections in particular.

# Materials and Methods

### Cells

Calu-3 human airway epithelial cells (HTB55; ATCC) were cultured in DMEM-Ham F-12 medium (1:1) (Gibco) supplemented with 10% FCS, penicillin, streptomycin, and glutamine, with fresh culture medium replenished every 2–3 d. Vero E6 (CRL-1586; ATCC) and HEK293 (CRL-1573; ATCC) cells were maintained in DMEM supplemented with 10% FCS, antibiotics, and glutamine.

### Virus and plasmids

Experiments with SARS-CoV-2 were performed under biosafety level 3 (BSL-3) conditions. The virus used in this study was SARS-CoV-2 isolate Munich 929 (kindly provided by Christian Drosten, Institute of Virology, Charité Universitätsmedizin Berlin). Virus stock was propagated on Vero E6 cells in DMEM medium with 1% FCS for 72 h. Cell supernatant was cleared by low-speed centrifugation and stored at –80°C.

The cDNA encoding the SARS-CoV-2 spike protein of isolate Wuhan-Hu-1 (GenBank accession number MN908947; codon-optimized, sequence available upon request) with a C-terminal Myc-6xHis-tag was synthesized at Eurofins and subcloned into in the pCAGGS expression plasmid using XhoI and NheI restriction sites (pCAGGS-S-Myc-6xHis). Expression plasmid pCAGGS-TMPRSS2 encoding the cDNA of human TMPRSS2 has been described previously (14).

### Antibodies

A polyclonal serum against 2002 SARS-CoV was generated by immunization of rabbits with inactivated SARS-CoV. A monoclonal mouse antibody against the C-terminal Myc-tag was purchased from Cell Signaling Technology (2276S). A monoclonal mouse anti-$\beta$ actin antibody was purchased from Abcam (ab6276). HRP-conjugated secondary antibodies were purchased from DAKO.

### PPMO

Phosphorodiamidate morpholino oligomers (PMOs) were synthesized at Gene Tools LLC. PMO sequences (5′–3′) were CAGAGTTG-GAGCACTTGCTGCCCA for T-ex5 and CCTCTTACCTCAGTTACAATTTATA for scramble. The cell-penetrating peptide (RXR)4 (where R is

arginine and X is 6-aminohexanoic acid) was covalently conjugated to the 3′ end of each PMO through a noncleavable linker, to produce peptide-PMO (PPMO), by methods described previously (62).

### Protease inhibitors

Aprotinin was purchased from AppliChem and the cysteine protease inhibitor E64d from Sigma-Aldrich (E8640). The synthetic inhibitors of TMPRSS2 and furin were synthesized in-house according to previous methods (29, 63). Stock solutions of protease inhibitors were prepared in double distilled water (aprotinin, MI-432, and MI-1851) or sterile DMSO (MI-1900 and E64d) and stored at –20°C.

### Synthesis of FRET substrates

The peptides were synthesized by automated solid phase peptide synthesis on a Syro 2000 synthesizer (MultiSynTech GmbH) using ~100 mg Rink-amide-MBHA resin (loading 0.68 mmol/g) for each 2 ml reaction vessel and a standard Fmoc-protocol with double couplings (approximately fourfold excess of Fmoc amino acid, HOBt and HBTU, respectively, and 8 equiv. DIPEA, 2 × 2 h coupling time) as described recently (64). After final coupling of Boc-2-aminobenzoic acid, the resin was washed with 20% piperidine in DMF (5 and 15 min) to remove an acylation on the 3-nitrotyrosine (65). The peptides were cleaved from the resin and deprotected by a mixture of TFA/triisopropylsilane/water (95/2.5/2.5, vol/vol/v) over 2 h at RT, followed by precipitation in cold diethyl ether. All peptides were purified by preparative reversed-phase HPLC to more than 95% purity based on the detection at 220 nm and finally obtained as lyophilized TFA salts.

### Enzyme kinetic measurements with recombinant soluble human furin

The measurements were performed in black 96-well plates (Nunc) at RT with a microplate reader (Safire2, Tecan) at $\lambda_{ex}$ 320 nm and $\lambda_{em}$ 405 nm. Each well contained 20 $\mu$l of the substrate solution (dissolved in water) and 150 $\mu$l buffer (100 mM Hepes, 0.2% Triton X-100, 2 mM CaCl$_2$, 0.02% Natriumazid, and 1 mg/ml BSA, pH 7.0). The measurements were started by addition of 20 $\mu$l furin (66) solution (0.5 nM in assay). The measurements were performed for 5 min, and the steady-state rates were calculated from the slopes of the progress curves.

### RNA isolation, and RT-PCR analysis of exon skipping

For analysis of TMPRSS2-mRNA from PPMO-treated Calu-3 cells, cells were incubated with the indicated concentrations (25 $\mu$M) of T-ex5 or scramble PPMO or without PPMO in PPMO medium (DMEM supplemented with 0.1% BSA, antibiotics, and glutamine) for 24 h. Total RNA was isolated at the indicated time points using the RNeasy Mini Kit (QIAGEN) according to the manufacturer's protocol. RT-PCR was carried out with total RNA using the one-step RT-PCR kit (QIAGEN) according to the supplier's protocol. To analyze TMPRSS2-mRNAs for exon skipping, primers TMPRSS2-108fwd (5′-CTA CGA GGT GCA TCC-3′) and TMPRSS2-1336rev (5′-CCA GAG GCC CTC CAG CGT CAC CCT GGC AA-3′) designed to amplify a full-length PCR product of

1,228 bp from control cells and a shorter PCR fragment of about 1,100 bp lacking exon 5 from T-ex5–treated cells were used ([25]). RT-PCR products were resolved on a 0.8% agarose gel stained with ethidium bromide.

### Infection of cells and multicycle virus replication in the presence of protease inhibitors or PPMO

SARS-CoV-2 infection experiments of Calu-3 cells were performed in serum-free DMEM supplemented with glutamine and antibiotics (DMEM++). For analysis of multicycle replication kinetics Calu-3 cells were seeded in 12-well plates and grown to 90% confluence. Cells were then inoculated with virus at an MOI of 0.001 in DMEM++ for 1 h 30 min, washed with PBS, and incubated in DMEM supplemented with 3% FCS, glutamine, and antibiotics (DMEM+++) with or without addition of protease inhibitors or DMSO to the medium for 72 h. At 16, 24, 48, and 72 h postinfection (p.i.), supernatants were collected, and viral titers were determined by tissue culture infection dose 50% ($TCID_{50}$) titration as described below. In addition, cells were fixed and immunostained against viral proteins as described below at 72 h p.i. to evaluate virus spread and virus-induced CPE.

For PPMO treatment, Calu-3 cells were incubated with 25 $\mu$M T-ex5 or scramble PPMO or remained untreated in PPMO medium for 24 h before infection. Cells were infected as described above and incubated in DMEM+++ without PPMO for 72 h.

### Virus titration by $TCID_{50}$

Viral supernatants were serial diluted in DMEM++. Each infection time point was titrated in four replicates from $5^1$ to $5^{11}$. Subsequently, 100 $\mu$l of each virus dilution were transferred to Calu-3 cells grown in 96-well plates containing 100 $\mu$l DMEM+++ and incubated for 72 h. Viral titers were determined with Spearman and Kärber algorithm described in reference [67].

### Transient expression of SARS-CoV-2 S protein in HEK293 cells

For transient expression of SARS-CoV-2 S protein 60% confluent HEK293 cells were co-transfected with 1.6 $\mu$g of pCAGGS-S-Myc-6xHis and either 15 ng of empty pCAGGS vector or pCAGGS-TMPRSS2 using Liopfectamine 2000 (Invitrogen) according to the manufacturers protocol for 48 h. Cells were harvested and centrifuged for 5 min at 8,000$g$. Subsequently, cells were subjected to SDS–PAGE and Western blot analysis as described below.

### SDS–PAGE and Western blot analysis

Cells were washed with PBS, lysed in CelLytic M buffer (Sigma-Aldrich) with a protease inhibitor cocktail (P8340; Sigma-Aldrich), resuspended in reducing SDS–PAGE sample buffer, and heated at 95°C for 10 min. Proteins were subjected to SDS–PAGE (10% acrylamide gel), transferred to a polyvinylidene difluoride (PVDF) membrane (GE Healthcare), and detected by incubation with primary antibodies and species-specific peroxidase-conjugated secondary antibodies. Proteins were visualized using the ChemiDoc XRS system with Image Lab software (Bio-Rad).

### Immunohistochemical staining and microscopy

To visualize viral spread in SARS-CoV-2–infected Calu-3 cells, immunohistochemical staining was performed. Calu-3 cells were fixed 72 h postinfection in 4% PFA for 36 h at 4°C. The cells were permeabilized with 0.3% Triton X-100 (Sigma-Aldrich) for 20 min at RT. The cells were incubated with a polyclonal rabbit serum against 2002 SARS-CoV for 1 h 30 min at RT, a species-specific peroxidase-conjugated secondary antibody for 1 h at RT and subsequently stained using the peroxidase substrate KPL TrueBlue (Seracare) and further analyzed on a Leica Dmi1 microscope.

### Cell viability assay

Cell viability was assessed by measuring the cellular ATP content using the CellTiterGlo luminescent cell viability assay (Promega). Calu-3 cells grown in 96-well plates were incubated with 25 $\mu$M of each PPMO or 50 $\mu$M of each of the protease inhibitors for 24 and 72 h, respectively. Subsequently, cells were incubated with the substrate according to the manufacturer's protocol. Luminescence was measured using a 96-well plate (Nunc) with a luminometer (Centro LB 960; Berthold Technologies). The absorbance values of PPMO- or inhibitor-treated cells were converted to percentages by comparison with untreated control cells, which were set at 100% cell viability.

## Data Availability

Data supporting the findings of this study are available within the article, its supplementary materials, or if stated otherwise available from the corresponding author (Eva Böttcher-Friebertshäuser, friebertshaeuser@staff.uni-marburg.de) upon reasonable request.

## Supplementary Information

## Acknowledgements

We are grateful to Christian Drosten for providing the virus. We thank Iris Lindberg for providing recombinant furin. We also thank Stephan Becker for support of this study. This work was funded by the LOEWE Center DRUID (project D1), by the German Center for Infection Research (DZIF), and by the Deutsche Forschungsgemeinschaft (DFG, German Research Foundation) SFB 1021 (project B07). We thank Diana Kruhl and Dirk Becker for excellent technical assistance.

### Author Contributions

D Bestle: data curation, formal analysis, validation, investigation, methodology, and writing—original draft, review, and editing.
MR Heindl: data curation, formal analysis, validation, investigation, methodology, and writing—original draft, review, and editing.

H Limburg: data curation, formal analysis, validation, investigation, methodology, and writing—original draft, review, and editing.

T Van Lam van: data curation, formal analysis, investigation, and methodology.

O Pilgram: data curation, formal analysis, investigation, and methodology.

H Moulton: conceptualization, resources, data curation, formal analysis, investigation, and methodology.

DA Stein: conceptualization, resources, and methodology.

K Hardes: conceptualization, resources, formal analysis, and methodology.

M Eickmann: conceptualization, resources, formal analysis, and methodology.

O Dolnik: resources and methodology.

C Rohde: resources and methodology.

H-D Klenk: conceptualization, resources, methodology, and writing—review and editing.

W Garten: conceptualization and writing—review and editing.

T Steinmetzer: conceptualization, resources, data curation, formal analysis, supervision, funding acquisition, and writing—original draft, review, and editing.

E Böttcher-Friebertshäuser: conceptualization, resources, data curation, formal analysis, supervision, funding acquisition, validation, investigation, methodology, project administration, and writing—original draft, review, and editing.

## Conflict of Interest Statement

The authors declare that they have no conflict of interest.

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
