## [Reviewer comments · Life Science Alliance]

Life Science Alliance

TMPRSS2 and furin are both essential for proteolytic activation of SARS-CoV-2 in human airway cells

Dorothea Bestle, Miriam Heindl, Hannah Limburg, Thuy Van Lam van, Oliver Pilgram, Hong Moulton, David Stein, Kornelia Hardes, Markus Eickmann, Olga Dolnik, Cornelius Rohde, Hans Klenk, Wolfgang Garten, Torsten Steinmetzer, and Eva Böttcher-Friebertshäuser

DOI: <https://doi.org/10.26508/lsa.202000786>

Corresponding author(s): Eva Böttcher-Friebertshäuser, Philipps-Universität Marburg

Review Timeline:	Submission Date:	2020-05-20
	Editorial Decision:	2020-06-03
	Revision Received:	2020-07-10
	Editorial Decision:	2020-07-14
	Revision Received:	2020-07-15
	Accepted:	2020-07-15

Transaction Report:

June 3, 2020

Re: Life Science Alliance manuscript #LSA-2020-00786-T

Prof. Eva Böttcher-Friebertshäuser
Philipps-University Marburg
Institute of Virology
Hans-Meerwein-Str. 2
Marburg 35043
GERMANY

Dear Dr. Böttcher-Friebertshäuser,

Thank you for submitting your manuscript entitled "TMPRSS2 and furin are both essential for activation and spread of SARS-CoV-2 in human airway cells" to Life Science Alliance. The manuscript was assessed by expert reviewers, whose comments are appended to this letter.

The referees appreciate the findings and have noted a few minor revisions that need to be sorted out. You can use the link below to upload the revised version.

Thank you for this interesting contribution to Life Science Alliance. We are looking forward to receiving your revised manuscript.

Sincerely,

Reilly Lorenz
Editorial Office Life Science Alliance

Meyerhofstr. 1
69117 Heidelberg, Germany
t +49 6221 8891 414
e contact@life-science-alliance.org
www.life-science-alliance.org

B. MANUSCRIPT ORGANIZATION AND FORMATTING:

Reviewer #1 (Comments to the Authors (Required)):

This is a superb paper describing the specificity of the furin and TMPRSS2 cleavage sites of the SARS-CoV-2 spike protein. The data using synthetic furin substrates show that the RRAR sequence in the spike protein is efficiently cleaved by furin. The authors make good use of different cell lines showing that furin processes the S1/S2 site and TMPRSS2 the S2' site. They further show that TMPRSS2 when down regulated blocks SARS-CoV-2 replication. Definitely the authors show that both TMPRSS2 and furin are critical for S activation. The authors then provide good evidence that protease inhibitors could be promising therapeutics. Three points:

- 1) There are no animal data provided. This should be discussed!
 - 2) It is not clear how the MI inhibitors were obtained. Please give details as to the availability (if any) of these compounds.
 - 3) Reading the manuscript one gets the impression that a therapeutic based on protease inhibition is around the corner. Aprotinin can be a highly toxic compound. It has been taken off the market! A balanced discussion might be appropriate!!!
- Overall this is an excellent paper!

Reviewer #2 (Comments to the Authors (Required)):

In the work "TMPRSS2 and furin are both essential for proteolytic activation and spread of SARS-CoV-2 in human airway cells and provide promising drug targets," Bestle et al describe the cleavage of the human coronavirus SARS-CoV-2 spike (S) protein by host proteases furin and TMPRSS2. As noted in the introduction, S must be cleaved at two sites to facilitate viral entry: S1/2 and S2'. The main points of the paper are as follows:

1.) The authors demonstrate with FRET substrates that furin is active on the S1/S2 site of SARS-CoV-2 and IBV, but not SARS-CoV and MERS-CoV. Using HEK293 cells, the authors suggest the S1/S2 is cleaved by the endogenous furin, whereas the S2' site is cleaved by TMPRSS2 when provided via transfection. In the presence of the furin inhibitor MI-1851, no band for S2 is visible on the blot. When TMPRSS2 is included, an additional band identified as S2' is visible.

Comments:

The S1 and S2 subunits of S are very close in size, and the recombinant protein subunits that are commercially available are advertised to run at about 80kDa. Of course, glycosylation states and tags would change this size, but have the authors confirmed the identity of the bands on their gel? Is it feasible that the bands they are calling S2 and S2' are S1 and S2? If not, where is S1?

The authors do address the combined impact of furin and TMPRSS2 with the inclusion or exclusion of the furin inhibitor, however, they don't seem to include aprotinin in combination with TMPRSS2 co-transfection experiments. This should be an easy experiment to do since they include it in the control and in follow-up experiments, and it would, in theory, result in the loss of the S2' band, corroborating their claims.

2.) The authors demonstrate that TMPRSS2 expression is required for SARS-CoV-2 replication in Calu-3 cells. They employ a PPMO specifically targeted to TMPRSS2 and show via analysis of mRNA that they are able to knockdown levels of the full-length mRNA, reducing the functional TMPRSS2 in the cell. They also confirm, via cell viability assays, that there are no adverse effects on cell growth. Viral growth is inhibited in the early stages of infection (16-48 hpi) in the TMPRSS2 knockdown cells confirming that TMPRSS2 is important for early replication.

Comments:

Consider changing the label on 3A to untreated instead of w/o for clarity.

In the figure legend for B, the text is "Calu-3 cells.... then infected with SARS-CoV-2 for 72 h as described above." This statement should be reworded for clarity, as it can be interpreted that the cells were in the presence of the inoculum for 72 hours, whereas you describe the inoculation lasting for 1.5 hours previously.

3.) The authors examine the effects of furin and TMPRSS2 inhibitors on SARS-CoV-2 replication in Calu-3 cells. They characterize the cytopathic effects, and complete growth curve analysis in concentration dependent conditions. Finally, they determine the synergistic effects of combining the furin and TMPRSS2 inhibitors. They conclude that inhibiting either protease stalls virus replication and as such, both are required. They also show that, when combined, the inhibitors have synergistic effects allowing for a smaller dosage of each to be effective.

Comments:

The immunostaining images seem to suggest that at 72 hpi, the inhibitors at concentrations of 20uM or greater have a significant impact on viral replication, however the TCID50 values indicate that by that time point, viral replication is nearly at the same level regardless of presence or absence of inhibitor. Do the authors have an explanation for this discrepancy?

For Figure 4, the images are very washed out and hard to interpret beyond the most basic recognition of some blue coloration. Higher contrast pictures or even those with high magnification would be easier to interpret. Also, it would be useful in the figures with images to denote what constitutes CPE and also point out examples of the punctate staining mentioned.

In Figure 4, the 20uM panel for MI-1900 and MI-1851 appear to be the same image.

In lines 297-298, the authors state that "aprotinin suppressed virus replication 25- to 100-fold compared to control cells even at a concentration of 10uM." While a significant reduction in titer is seen at early time-points, by the end of 72 hours the difference does not appear to be significant. A more appropriate statement might highlight the early effects of the inhibitor, as this statement seems to suggest that the effectiveness is not correlated with the time post-infection. Likewise, this waning effect of the inhibitors is not addressed throughout the rest of the paper, even into the arguments in the discussion for why this may be a good therapeutic. In reality, how feasible would it be to give a drug 16-24 hours after infection? These experiments all seem to be set up with the inclusion of the inhibitor only at the time of inoculation. Have the authors tried to replenish the inhibitor to see if the titers are kept low at later time-points post-infection?

For the combination treatments, in all cases while combining inhibitors does lower the viral titer compared to uninhibited controls, there is still appreciable growth. If both proteases are required for S- activation, can the authors speculate on why virus is detectable and amplifies from 16 hpi on? Is it simply a matter of incomplete inhibition (could be suggested by the result in 5D with the TMPRSS2 knockdown)?

Lines 348 and 350-351 seem to overstate the findings of the inhibitor experiments. Please consider revising. Even at the first time point in the combined inhibitor experiment, virus is detectable suggesting that though growth was slowed, the activity of the protease was not "essential" for replication. Similarly, the claim that "TMPRSS2 and furin cannot compensate for each other in SARS-CoV-2 activation" does not seem to be supported by the results in Fig 5a, showing that virus is able to replicate and grow, even if at a slower rate, in the presence of one type of inhibitor or the other.

Minor Comments:

Consider adding a label for figure 6 denoting the S1/S2 and S2' cleavage sites on the diagram.

Overall, nice work!

Reviewer #3 (Comments to the Authors (Required)):

This manuscript describes the use of specific peptidomimetic and PPMO inhibitors to confirm that both furin and TMPRSS2 cleavages of the SARS-CoV2 S protein precursor are required for virus replication, and that inhibitors to the respective proteases act synergistic to limit SARS-CoV2 replication in airway epithelial cell culture. The manuscript is well written and the experimental results are clear and unambiguous. Although the work leaves open some questions - is furin cleavage a prerequisite for TMPRSS2 cleavage? is the cleavage by endosomal cathepsins reported by others in fact a fallback adaptation to growth in cell culture? do alternative genetic adaptations enable S1/S2 cleavage/fusion in CoVs that lack a furin cleavage site? does a 2-protease process enable stricter control of S protein fusogenicity than a 1-enzyme process? - it does provide a full and comprehensive description of what is likely the primary pathway of SARS-CoV2 entry and suggests sound intervention strategies.

As noted, the manuscript is well written and tightly reasoned. One minor (and inconsequential) question arises from experiments examining S protein cleavage in HEK cells that have been co-transfected to express TMPRSS2, which is otherwise absent in these cells (Fig. 2C). The extent of S2' cleavage is extremely low (and seemingly unchanged regardless of furin levels), thereby raising questions as to the validity of the co-transfection protocol. Subsequent studies with a specific TMPRSS2-targeting PPMO in TMPRSS2-expressing Calu-2 cells definitively confirm the importance of TMPRSS2 cleavage for SARS-CoV2 replication.

Thank you very much for editing our manuscript „TMPRSS2 and furin are both essential for proteolytic activation and spread of SARS-CoV-2 in human airway cells and provide promising drug targets“ by Bestle et al. (LSA-2020-00786-T). We are pleased to submit our revised manuscript.

We are glad and grateful for the very positive response to our manuscript from all review-ers. The reviewers provided several helpful comments and suggestions for the improve-ment of our manuscript and we used the comments to guide the production of our revised manuscript.

Among other improvements, the revised manuscript contains additional data over that present in the first submission in Fig. 2C.

Attached to this letter is a point by point response to the reviewer's comments, which we hope will help to render our manuscript acceptable for publication in Life Science Alliance.

Reviewer 1:

1) There are no animal data provided. This should be discussed!

Of course it will be very interesting to examine the antiviral efficacy of the different protease inhibitors against SARS-CoV-2 in an animal model. However, it still takes some time until we can start with infection experiments in animals in vivo for several reasons:

Inhibitors MI-1900 and MI-1851 were developed by us very recently and we just started to examine toxicity and pharmacokinetic and pharmacodynamic (PK/PD) parameters in mice and rats as a first step towards testing their antiviral efficacy against SARS-CoV-2 as well as other respiratory viruses in an animal model.

In addition, the development of a suitable mouse model for SARS-CoV-2 is still ongoing at our institute. Alternatively, to order mice with human ACE2 expression is very difficult at the moment due to the high demand worldwide. We may have the option to test the inhibitors in the Syrian hamster model in collaboration with another lab in the near future, however, we currently wait for the results from the toxicity and PK/PD studies.

The role of TMPRSS2 in SARS-CoV-2 activation could be investigated in TMPRSS2-deficient mice. We breed the mice at our institute, but again we would have to establish TMPRSS2-deficient mice expressing human ACE2. This can be performed by adenoviral transduction.

The development of PPMO targeting host factors as antiviral therapy has one disadvantage: the target RNA sequence differs in different host species and a new PPMO has to be developed in order to test the antiviral efficacy of a PPMO that works well in human cells in an animal model. Thus, we have to develop a new PPMO analogous to T-ex5 that is complementary to the sequence of either mouse or hamster TMPRSS2-pre-mRNA at the intron/exon junction of exon 5. Human, mouse and hamster TMPRSS2 pre-mRNA sequences, respectively, show several mismatches at the i5e5 junction. In addition, it has to be investigated whether the new PPMO causes efficient exon skipping also in hamster or mouse TMPRSS2 pre-mRNA or whether another mRNA sequence provides a better target in that case. One solution would be development of dual PPMO that fit to both sequences and show 1-2 mismatches for each sequence. We are currently testing such a dual mouse-human T-ex5 analogous PPMO in human and murine airway cells.

2) It is not clear how the MI inhibitors were obtained. Please give details as to the availability (if any) of these compounds.

MI-432, MI-1851 and MI-1900 were synthesized in-house by the working group of Torsten Steinmetzer (Thuy van Lam Van, Oliver Pilgram and Torsten Steinmetzer are co-authors of the manuscript). We added a comment on that in the material and methods section (line 563).

The synthesis of MI-432, MI-1900 and MI-1851 is described in the results section and material and methods section (see lines 161-162, 250-256 and 563-564). Synthesis of MI-432 is described in reference 28, MI-1851 was synthesized according to protocols described in reference 62. The synthesis of MI-1851 is described in detail in a manuscript

that has been submitted by us to MedChemLetters very recently. We added a comment on that in the manuscript (line 161). The inhibitor MI-1900 was prepared by an analogous strategy as described for MI-432, but has not been published so far. All inhibitors are available upon request from the corresponding author or from Torsten Steinmetzer.

3) *Reading the manuscript one gets the impression that a therapeutic based on protease inhibition is around the corner. Aprotinin can be a highly toxic compound. It has been taken off the market! A balanced discussion might be appropriate!!!*

It is correct that the benefit-risk balance of aprotinin has been discussed controversial within the last decade. Aprotinin given as injection has been widely used to reduce bleeding during cardiac and liver surgeries. Aprotinin (Trasylol, BAYER) was temporarily withdrawn worldwide in 2007 after studies suggested an increased risk of complications or death after usage during complex surgeries. In 2011/2012 Health Canada and the European Medicines Agency (EMA) Committee for Medicinal Products re-analysed the Trasylol studies from 2006-2008 and determined that the benefits of aprotinin outweigh the risks when used for patients undergoing coronary artery bypass graft surgery and that the evidence does not suggest an increased risk of death in this context. In 2012 the EMA recommended to the EU that the suspension of the licence for aprotinin in the context of coronary artery bypass graft surgery be lifted. Nordic Pharma acquired the rights to Trasylol outside of the USA in 2012.

Studies by Oleg Zhirnov and colleagues have demonstrated that aerosol inhalations of aprotinin reduce the duration of symptoms in influenza patients without causing side effects. Thus, aprotinin may provide a therapeutic approach also for other respiratory viruses that require TMPRSS2 for proteolytic activation. Moreover, studies by Zhirnov and co-workers suggest that aerosolized aprotinin may provide a promising prophylactic approach for influenza. Although it has not been studied in detail so far the inhalation of aprotinin most likely has a lower risk of side effects compared to injection. Due to the urgent need of therapeutic treatments for COVID-19 and the rationale of drug repurposing aprotinin inhalation provides a promising approach that should at least be tested in animal models.

Reviewer 2:

1) The S1 and S2 subunits of S are very close in size, and the recombinant protein subunits that are commercially available are advertised to run at about 80kDa. Of course, glycosylation states and tags would change this size, but have the authors confirmed the identity of the bands on their gel? Is it feasible that the bands they are calling S2 and S2' are S1 and S2? If not, where is S1?

We used antibodies against the C-terminal Myc epitope of the S protein expressed by our expression plasmid. Therefore, we can only detect the subunits with C-terminal Myc-tag by Western blot analysis (S2 and S2'). The S1 subunit lacks a Myc tag and cannot be detected in our case. For clarity, we added a comment on that in the main text (lines 166-167).

2) *The authors do address the combined impact of furin and TMPRSS2 with the inclusion or exclusion of the furin inhibitor, however, they don't seem to include aprotinin in combination with TMPRSS2 co-transfection experiments. This should be an easy experiment to do since they include it in the control and in follow-up experiments, and it would, in theory, result in the loss of the S2' band, corroborating their claims.*

Thank you for this comment. We performed the experiment. As expected, the amount of S2' protein is strongly reduced in the presence of aprotinin. We show the data in the right panel of Fig. 2C and revised the main text according to the data.

3) *Consider changing the label on 3A to untreated instead of w/o for clarity.*

We decided to keep w/o for without treatment instead of untreated, since we prefer the shorter term. However, for clarity we reworded untreated by "without treatment" in figure legend 3A.

4) *In the figure legend for B, the text is "Calu-3 cells... then infected with SARS-CoV-2 for 72 h as described above." This statement should be reworded for clarity, as it can be interpreted that the cells were in the presence of the inoculum for 72 hours, whereas you describe the inoculation lasting for 1.5 hours previously.*

We reworded the statement in the figure legend according to the suggestion (highlighted in yellow).

5) *The authors examine the effects of furin and TMPRSS2 inhibitors on SARS-CoV-2 replication in Calu-3 cells. They characterize the cytopathic effects, and complete growth curve analysis in concentration dependent conditions. ... The immunostaining images seem to suggest that at 72 hpi, the inhibitors at concentrations of 20uM or greater have a significant impact on viral replication, however the TCID50 values indicate that by that time point, viral replication is nearly at the same level regardless of presence or absence of inhibitor. Do the authors have an explanation for this discrepancy?*

In protease inhibitor treated cells the CPE is delayed due to delayed virus replication and increase in virus titer. For instance, in untreated control cells or E64d treated cells a strong CPE is visible at 72 h p.i. consistent with high virus titers already at 24 h p.i.. In cells treated with aprotinin or MI-432 virus titers are strongly reduced at early time points and then increase at 48 h. Thus, CPE due to high viral titers is visible at later time points compared to control cells. In cells treated with MI-1900 we see a dose-dependent CPE at 72 h p.i. consistent with virus titers. In the presence of 10 uM MI-1900 we see a clear CPE at 72 h p.i. in agreement with virus titers similar to control cells at 72 h p.i. In contrast, in cells treated with 50 uM MI-1900 virus titers are still markedly reduced at 72 h p.i. and also virus induced CPE is not observed yet. In MI-1851 treated cells we observe strong reduction of virus replication at all time points and this is in agreement with the images.

In order to show the correlation of CPE and reduction in virus titers more clearly we substituted the image of 10 uM MI-432 by a more representative image (from four independent experiments). Here, virus spread is clearly visible at 72 h p.i. consistent with an increase in virus titers already at 48 h p.i.

6) *For Figure 4, the images are very washed out and hard to interpret beyond the most basic recognition of some blue coloration. Higher contrast pictures or even those with high magnification would be easier to interpret. Also, it would be useful in the figures with images to denote what constitutes CPE and also point out examples of the punctate staining mentioned.*

We agree with the reviewer that the staining of some of the images is a bit washed out. We used the peroxidase-substrate TrueBlue for IHC staining of virus-positive cells. The substrate is water soluble and sometimes the staining intensity decreases and some of the infected cells are stained only light blue or grey instead of dark blue. In addition, the SARS-CoV-2 infected cells have to be fixed for 36 h with PFA to ensure virus inactivation and to perform immunostaining and microscopy outside the BSL3 lab. This often results in weaker staining of virus-positive cells by our SARS-CoV serum.

To improve the figure, we put together smaller sections of the original images with higher magnification as suggested by the reviewer. The virus-induced CPE (holes visible throughout the cell monolayer), virus-positive cells stained in blue and foci of infection are much easier to interpret now. In addition, we explained the CPE and blue staining of virus-positive cells more clearly in the main text and figure legends (highlighted in yellow).

7) *In Figure 4, the 20uM panel for MI-1900 and MI-1851 appear to be the same image.*

Thank you very much for this comment! This is correct and we very much apologize that we have overseen this mistake during preparation and proof reading of the manuscript. We substituted the image for 20 uM MI-1900 by the correct image.

8) *In lines 297-298, the authors state that "aprotinin suppressed virus replication 25- to 100-fold compared to control cells even at a concentration of 10uM." While a significant reduction in titer is seen at early time-points, by the end of 72 hours the difference does not appear to be significant. A more appropriate statement might highlight the early effects of the inhibitor, as this statement seems to suggest that the effectiveness is not correlated with the time post-infection. Likewise, this waning effect of the inhibitors is not addressed throughout the rest of the paper, even into the arguments in the discussion for why this may be a good therapeutic. In reality, how feasible would it be to give a drug 16-24 hours after infection? These experiments all seem to be set up with the inclusion of the inhibitor only at the time of inoculation. Have the authors tried to replenish the inhibitor to see if the titers are kept low at later time-points post-infection?*

This is an interesting and important comment and we added few comments on that in the discussion section (442-447). We think that the increase in virus titers in aprotinin and MI-432 treated cells is indeed due to waning inhibitor efficacy at later time points and proba-

bly can be overcome by repeated inhibitor treatment at 24 or 48 h post infection. In cells treated with 50 uM of either MI-1851 or MI-1900 final titers are still markedly reduced compared to untreated cells also at 72 h p.i.. We have not investigated the effect of repeated inhibitor treatment of SARS-CoV-2 infected Calu-3 cells nor the stability/half-life of the different protease inhibitors in cells so far, but we plan to do it in future studies. For treatment of COVID-19 inhibitors most likely would be administered several times per day.

9) *For the combination treatments, in all cases while combining inhibitors does lower the viral titer compared to uninhibited controls, there is still appreciable growth. If both proteases are required for S- activation, can the authors speculate on why virus is detectable and amplifies from 16 hpi on? Is it simply a matter of incomplete inhibition (could be suggested by the result in 5D with the TMPRSS2 knockdown)?*

Yes, we think that this is simply a matter of incomplete inhibition of S cleavage. However, e.g. the combination of 50 uM each MI-1851 and MI-432 shows drastically reduction of virus replication and might cause complete blockage of virus activation and multiplication by repeated inhibitor treatment or pre-treatment of cells prior to infection in order to ensure that furin and TMPRSS2, respectively, are efficiently inhibited already during the first replication cycle.

Complete inhibition of S cleavage may be not so trivial. It is not clear what ratio of fully cleaved S versus uncleaved/incompletely cleaved S on the virus envelope is sufficient to facilitate virus entry. Maybe a patch of few fully processed spike proteins is already sufficient to mediate membrane fusion.

However, our data show that inhibition of S cleavage by host cell protease inhibitors strongly suppresses SARS-CoV-2 replication. A combination of host cell protease inhibitors and antiviral drugs may show synergistic effects and provide a promising approach for COVID-19 treatment. We discussed this in the discussion section.

10) *Lines 348 and 350-351 seem to overstate the findings of the inhibitor experiments. Please consider revising. Even at the first time point in the combined inhibitor experiment, virus is detectable suggesting that though growth was slowed, the activity of the protease was not "essential" for replication. Similarly, the claim that "TMPRSS2 and furin cannot compensate for each other in SARS-CoV-2 activation" does not seem to be supported by the results in Fig 5a, showing that virus is able to replicate and grow, even if at a slower rate, in the presence of one type of inhibitor or the other.*

We toned down the conclusions (highlighted in yellow).

344-345: " ... both TMPRSS2 and furin are essential for efficient virus multicycle replication in Calu-3 human airway cells."

347: "The results suggest that TMPRSS2 and furin cannot compensate for each other in SARS-CoV-2 S activation."

Minor Comments:

Consider adding a label for figure 6 denoting the S1/S2 and S2' cleavage sites on the diagram.

We added a label for S1/S2 and S2' cleavage sites on the diagram and also highlighted the "uncleaved" sites by an additional line to improve the figure. Thank you for the suggestion - it improved the scheme.

Reviewer 3:

The manuscript is well written and the experimental results are clear and unambiguous. Although the work leaves open some questions - is furin cleavage a prerequisite for TMPRSS2 cleavage? is the cleavage by endosomal cathepsins reported by others in fact a fallback adaptation to growth in cell culture? do alternative genetic adaptations enable S1/S2 cleavage/fusion in CoVs that lack a furin cleavage site? does a 2-protease process enable stricter control of S protein fusogenicity than a 1-enzyme process? - it does provide a full and comprehensive description of what is likely the primary pathway of SARS-CoV2 entry and suggests sound intervention strategies.

Thank you for all the interesting comments and thoughts! Indeed, these are important questions that are still open and keep many labs busy.

1) *One minor (and inconsequential) question arises from experiments examining S protein cleavage in HEK cells that have been co-transfected to express TMPRSS2, which is otherwise absent in these cells (Fig. 2C). The extent of S2' cleavage is extremely low (and seemingly unchanged regardless of furin levels), thereby raising questions as to the validity of the co-transfection protocol. Subsequent studies with a specific TMPRSS2-targeting PPMO in TMPRSS2-expressing Calu-2 cells definitively confirm the importance of TMPRSS2 cleavage for SARS-CoV2 replication.*

In our transfection experiment we used 100fold less TMPRSS2 expression plasmid compared to S expression plasmid for transfection in order to analyse S cleavage by TMPRSS2 (described in material and methods). This is a protocol we have established for co-expression of influenza virus HA and TMPRSS2. For so far unknown reasons expression of the viral fusion protein is reduced by expression of high levels of TMPRSS2, expression of TMPRSS2 seems to be preferred by the cells. This has also been observed by other labs. That's why we always use lower amounts of TMPRSS2 compared to the viral fusion protein in co-expression experiments. Of course, this results in lower levels of TMPRSS2 activity and, consequently, the amount of S2' is very low. We have repeated the experiment 4 times with the same results and low amount of S2'. However, we think although the amount of S2' is very low the co-expression experiment shows that cleavage of S by furin and TMPRSS2 occurs at different cleavage sites. Evidence for the importance of TMPRSS2 in SARS-CoV-2 activation and replication is provided by the strong reduction of SARS-CoV-2 replication in Calu-3 cells upon PPMO-induced knockdown of TMPRSS2 expression.

July 14, 2020

RE: Life Science Alliance Manuscript #LSA-2020-00786-TR

Prof. Eva Böttcher-Friebertshäuser
Philipps-Universität Marburg
Institute of Virology
Hans-Meerwein-Str. 2
Marburg 35043
Germany

Dear Dr. Böttcher-Friebertshäuser,

Thank you for submitting your revised manuscript entitled "TMPRSS2 and furin are both essential for proteolytic activation of SARS-CoV-2 in human airway cells". We would be happy to publish your paper in Life Science Alliance pending final revisions necessary to meet our formatting guidelines.

-please add ORCID ID for corresponding author-you should have received instructions on how to do so

-Thank you for indicating the splice sites in Fig. 2C and Fig. 3C. We would recommend using either a blank space or a black line to more clearly indicate that the panels were assembled with different blots, and kindly ask you to make this change.

-please provide source data for Fig. 2C & Fig. 3C

-please add scale bars to Fig. 3A & Fig. 4

A. FINAL FILES:

-- Summary blurb (enter in submission system): A short text summarizing in a single sentence the

study (max. 200 characters including spaces). This text is used in conjunction with the titles of papers, hence should be informative and complementary to the title. It should describe the context and significance of the findings for a general readership; it should be written in the present tense and refer to the work in the third person. Author names should not be mentioned.

B. MANUSCRIPT ORGANIZATION AND FORMATTING:

Sincerely,

Reilly Lorenz
Editorial Office Life Science Alliance
Meyerhofstr. 1
69117 Heidelberg, Germany
t +49 6221 8891 414
e contact@life-science-alliance.org
www.life-science-alliance.org

July 15, 2020

RE: Life Science Alliance Manuscript #LSA-2020-00786-TRR

Prof. Eva Böttcher-Friebertshäuser
Philipps-Universität Marburg
Institute of Virology
Hans-Meerwein-Str. 2
Marburg 35043
Germany

Dear Dr. Böttcher-Friebertshäuser,

Thank you for submitting your Research Article entitled "TMPRSS2 and furin are both essential for proteolytic activation of SARS-CoV-2 in human airway cells". It is a pleasure to let you know that your manuscript is now accepted for publication in Life Science Alliance. Congratulations on this interesting work.

Once your paper is published, please alert BioRxiv to update publication status to 'Published at Life Science Alliance.'

DISTRIBUTION OF MATERIALS:

Again, congratulations on a very nice paper. I hope you found the review process to be constructive and are pleased with how the manuscript was handled editorially. We look forward to future exciting submissions from your lab.

Sincerely,

Reilly Lorenz
Editorial Office Life Science Alliance
Meyerhofstr. 1
69117 Heidelberg, Germany
t +49 6221 8891 414
e contact@life-science-alliance.org
www.life-science-alliance.org